# Microwave emission from superconducting vortices in Mo/Si superlattices

O.V. Dobrovolskiy [1,2], V.M. Bevz[2], M.Yu. Mikhailov [3], O.I. Yuzephovich[3], V.A. Shklovskij[2], R.V. Vovk[2], M.I. Tsindlekht[4], R. Sachser[1] & M. Huth [1]

Most of superconductors in a magnetic field are penetrated by a lattice of quantized flux vortices. In the presence of a transport current causing the vortices to cross sample edges, emission of electromagnetic waves is expected due to the continuity of tangential components of the fields at the surface. Yet, such a radiation has not been observed so far due to low radiated power levels and lacking coherence in the vortex motion. Here, we clearly evidence the emission of electromagnetic waves from vortices crossing the layers of a superconductor/insulator Mo/Si superlattice. The emission spectra consist of narrow harmonically related peaks which can be finely tuned in the GHz range by the dc bias current and, coarsely, by the in-plane magnetic field value. Our findings show that superconductor/insulator superlattices can act as dc-tunable microwave generators bridging the frequency gap between conventional radiofrequency oscillators and (sub-)terahertz generators relying upon the Josephson effect.

[1] Physikalisches Institut, Goethe University, Max-von-Laue-Str. 1, 60438 Frankfurt am Main, Germany. [2] Physics Department, V. N. Karazin Kharkiv National University, Svobody Square 4, Kharkiv 61022, Ukraine. [3] B. I. Verkin Institute for Low Temperature Physics and Engineering of the National Academy of Sciences of Ukraine, Nauky Avenue 47, Kharkiv 61103, Ukraine. [4] The Racah Institute of Physics, The Hebrew University of Jerusalem, Givat Ram, 91904 Jerusalem, Israel. Correspondence and requests for materials should be addressed to O.V.D. (email: dobrovolskiy@physik.uni-frankfurt.de)

In 1962, Josephson[1] predicted the emission of electromagnetic (em) radiation from superconducting tunnel junctions at the frequency $\omega = 2eV/\hbar$ determined only by the direct current (dc) voltage $V$ applied across the junction. Such a radiation into a waveguide at a power level of ~1 pW was detected by Yanson et al.[2] in 1964 and Langenberg et al.[3] in 1965. Josephson vortices, induced by an in-plane dc magnetic field, moved to the junction edge in the presence of the transport current and caused oscillations of the magnetic and electric fields. Later on, after the discovery of layered high-temperature superconductors, it has been recognized that Bi- and Tl-based cuprates with weakly coupled superconducting layers represent stacks of intrinsic Josephson junctions on the atomic scale[4]. Many Josephson junction-related effects have been observed in these systems, including terahertz emission[5,6], Shapiro steps in the $I–V$ curve induced by external microwave radiation[7,8], Fiske resonances[9], and oscillations of the critical current on the in-plane dc magnetic field. Current state-of-the-art Josephson flux-flow oscillators, which are used, e.g., in receivers in astrophysics, deliver micro-watt powers at frequencies of 400–600 GHz and free-running emission linewidths as low as 0.6 MHz[10–12]. Substantially higher emission powers can be obtained from one-dimensional or two-dimensional (2D) arrays of phase-locked Josephson junctions[13] with on-chip-detected emission powers approaching 0.2 mW[5,14].

A tunable model system for the study of the physics of layered superconductors has been established in artificial superlattices consisting of superconducting and non-superconducting layers in which the interlayer Josephson coupling can be tuned by varying the thickness of the interlayers and using a wide range of constituting materials[15–19]. As the interlayer coupling increases, a normal-core-free, Josephson phase vortex evolves into an Abrikosov vortex whose core is centered between the super-conducting layers[20–22]. By analogy with the radiation from a Josephson junction, possible em emission from a moving lattice of Abrikosov vortices has been pointed out by Kulik in a theoretical work back in 1966[23]. Namely, when a rather large external current is applied to a type-II superconductor, Abrikosov vortices move under the action of the Lorentz force and this vortex movement is accompanied by oscillations of the supercurrents and the associated magnetic induction[23,24]. Such supercurrent oscillations were experimentally observed in granular superconducting films by Martinoli et al.[25] and Hebboul et al.[26]. As the vortex lattice comes to a sample edge, the oscillating electric and magnetic fields of vortices should propagate into free space due to the continuity of tangential components of the fields at the surfaces[27,28]. The spectrum of the em radiation from the Abrikosov vortex lattice crossing a sample edge has been predicted to peak at the harmonics of the washboard frequency $f_0 = v/d$, where $v$ is the vortex velocity and $d$ is the distance between the vortex rows in the direction of motion[28]. Yet, such a radiation has not been observed so far, as its detection poses a severe experimental challenge. Namely, while the radiated power from a 1 mm$^2$ sample surface has been estimated to be of the order of ~$10^{-7}$ W for a triangular vortex lattice, this value drastically decreases in the presence of disorder[28]. While the vortex flow is known to become unstable at vortex velocities of the order of 1 km s$^{-1}$[29,30], for an em generation at $f_0 \simeq 10$ GHz, i.e., in the frequency range which is important for microwave applications, modulation of the em properties of the superconductor at a length scale of and below 100 nm is required. Unfortunately, a thin film geometry with an out-of-plane magnetic field makes an em generation detection barely feasible, as the area of the side surfaces crossed by vortices becomes negligibly small. In addition, uncorrelated disorder reduces the range of correlations in the vortex lattice, thus suppressing the radiated power levels even further.

Here we provide experimental evidence for the em radiation from a lattice of Abrikosov vortices moving across the layers in a superconductor/insulator Mo/Si superlattice. Emission powers at levels >$10^{-12}$ W are observed for a 5 mm$^2$ sample surface crossed by vortices in the coherent regime achieved at large matching values of the magnetic field when a dense vortex lattice is commensurate with the multilayer period. The emission is peaked at the harmonics of the washboard frequency $f_0$ that can be finely tuned from about 5 GHz to about 30 GHz by the dc bias current and, coarsely, by switching the in-plane magnetic field between matching values. Furthermore, by varying the size of the vortex cores by temperature, we exploit the dimensionality crossover of superconductivity in the superlattice for tailoring the emission spectra. Namely, we tune the frequency-selective em emission at the harmonics of $f_0$ related to the period of the vortex lattice crossing the sample edges to the harmonics of $2f_0$ related to the multilayer period when vortices fit in the insulating Si layers. Our findings show that superconductor/insulator multilayers can act as dc-tunable microwave generators bridging the frequency gap between conventional radio frequency (rf) oscillators and (sub-) terahertz generators relying upon the Josephson effect.

## Results

**Investigated system**. The investigated system is shown in Fig. 1a. The emission of em waves at microwave frequencies is detected from Abrikosov vortices crossing the layers in a superconductor/insulator Mo/Si superlattice. The superlattice consists of 50 alternatingly sputtered Mo and Si layers with thicknesses $d_{Mo} = 22$ Å and $d_{Si} = 28$ Å, resulting in a multilayer period $s$ of 50 Å. A transmission electron microscopic image of a part of the sample is shown in Fig. 1b. The superconducting transition temperature of the sample, determined at the midpoint of the resistive transition $R(T)$, is $T_c = 4.02$ K. The Josephson coupling between the superconducting Mo layers is rather strong, $\eta_J = \hbar^2/2ms^2\gamma^2 \approx 1$[15]. Here, $\hbar$ is the Planck constant, $m$ is the in-plane mass of the Cooper pairs, and $\gamma = 5.72$ is the anisotropy parameter. A four-probe $5 \times 1$ mm$^2$ bridge was patterned in the sample for electrical transport measurements, Fig. 1c. The magnetic field and transport current were applied in the layer plane and orthogonal to each other, causing a vortex motion across the layers under the action of the Lorentz force, Fig. 1a. The emitted signal was picked up by a small wire loop shorting the end of a semirigid coaxial cable and placed close to the sample surface[31]. The em emission was monitored by a high-frequency spectrum analyzer.

**Microwave emission from moving vortex lattice**. Figure 2 displays the emission spectra recorded at vortex velocities $v$ from 0 to 150 m s$^{-1}$ in magnetic fields $H = 3.15$ and 4.2 T at temperatures $T = 1.8$, 3, and 3.6 K. At $v = 0$, the noise floor of the detector is seen and there is no emission detected. At currents exceeding the depinning current, the vortices move across the layers. Since the depinning current decreases with increase of $T$ and $H$, in Fig. 3a we plot the $I–V$ curves in a vortex velocity $v$ versus normalized current $I/I^*$ representation. Here the vortex velocity $v$ was deduced from the $I–V$ curves using the standard relation $v = V/HL$, where $V$ is the measured voltage, $H$ is the magnetic field value, and $L = 2.44$ mm is the distance between the voltage contacts. The $I^*$ values were deduced for each of the $I–V$ curves at the intersect of the extrapolated linear section with the current axis, as exemplified in the inset of Fig. 3a. Accordingly, $I^*$ has the meaning of a depinning current determined by the "dynamic" criterion, as is commonly used for systems with strong pinning[20]. For vortex velocities between 50 and 150 m s$^{-1}$, the scaled $I–V$ curves in Fig. 3a fit to the universal relation $v = 109.5(I/I^* - 1)$ ms$^{-1}$ with an error of <5%. This relation not only allows for a direct

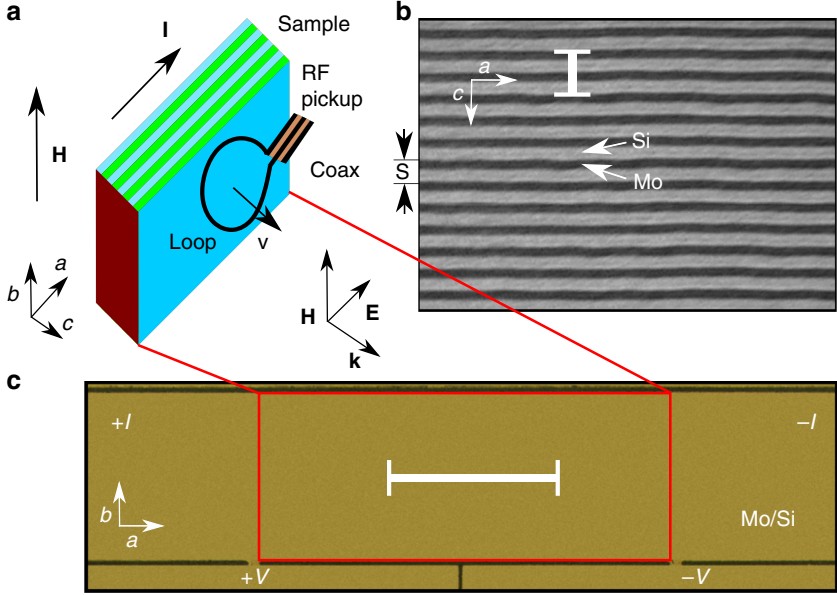

**Fig. 1** Superconductor/insulator Mo/Si superlattice. **a** Experimental geometry (not to scale). The Mo/Si multilayer is in a magnetic field **H** applied parallel to the $b$ axis. The transport current **I** applied along the $a$ axis causes the vortex lattice to move with velocity $v$ across the layers. Electromagnetic radiation from the lattice of flux lines crossing the superconducting layers is picked up by a wire loop antenna. **b** Transmission electron microscopic image of the Mo/Si multilayer with a multilayer period $s = d_{Mo} + d_{Si} = 50$ Å. The scale bar corresponds to 100 Å. **c** Optical microscopic image of the bridge etched in the Mo/Si multilayer. The scale bar corresponds to 1 cm

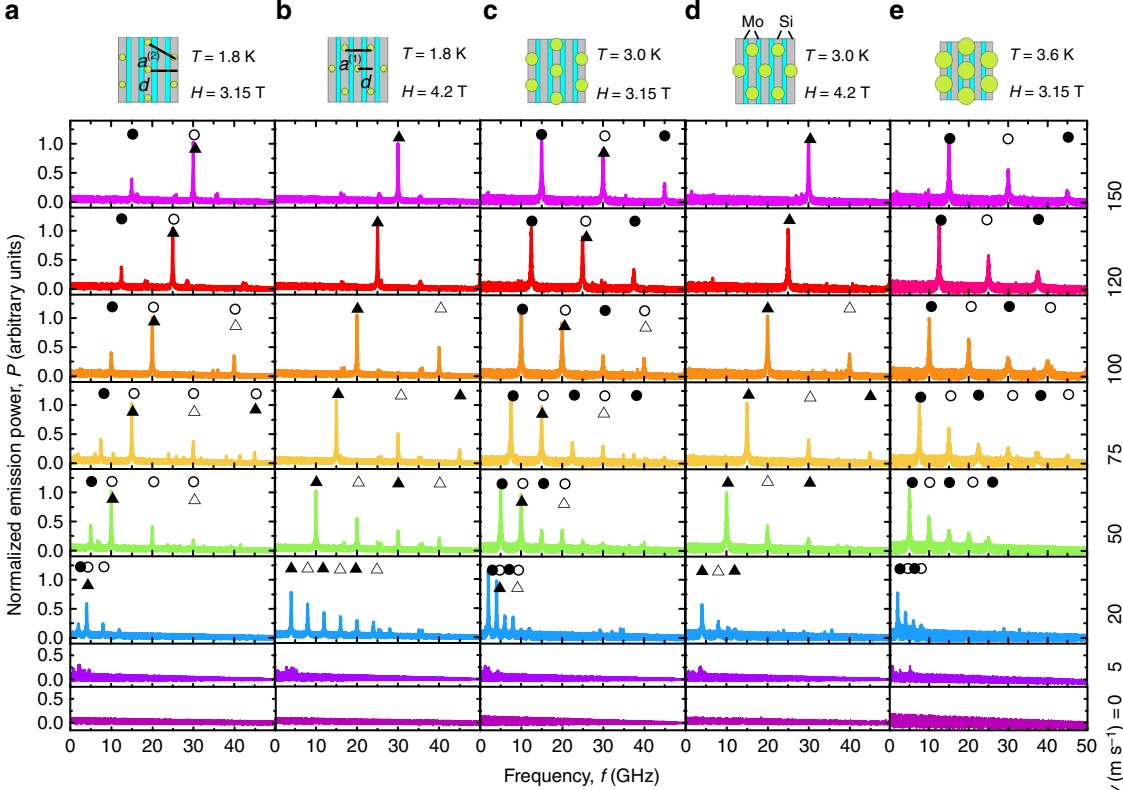

**Fig. 2** Microwave emission from Abrikosov vortices in the Mo/Si superlattice. **a–e** Emission spectra for a series of vortex velocities $v$, as indicated, at $H = 3.15$ and 4.2 T and $T = 1.8$, 3.0, and 3.6 K. The symbols above the emission peaks indicate the frequencies $f_m = mf_0$, which are harmonically related to the washboard frequency $f_0^{(1)}$ associated with the 50 Å-periodic layered structure (triangles) and $f_0^{(2)}$ related to the 100 Å-spaced vortex rows in the direction of their motion (circles). The odd and even harmonics are indicated by solid and open symbols, respectively. The vortex lattice configurations, which are commensurate with the Mo/Si multilayer period at $H_{N=2}^{(2)} = 3.15$ T with $a^{(2)} = 2d^{(2)}/\sqrt{3} = 4s/\sqrt{3}$ and $H_{N=1}^{(1)} = 4.2$ T with $a^{(1)} = 2d^{(1)} = 2s$, are shown in the scaled coordinate system $(\gamma a, c)$ above the panels. In the absence of transport current, vortices are centered in regions of reduced order parameter, the Si layers. The size of the circles denoting the vortex cores (not to scale) reflects their relation to the Si layer thickness $d_{Si}$ and the multilayer period $s$

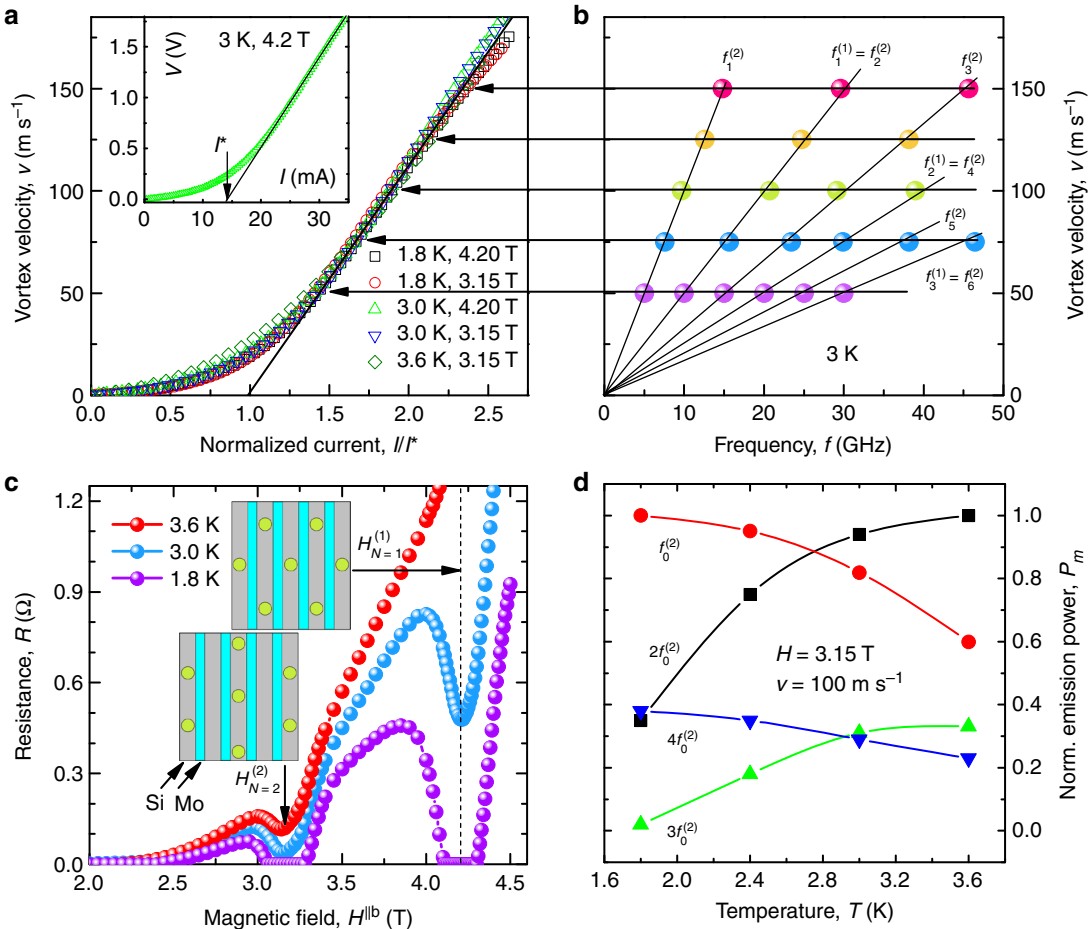

**Fig. 3** Emission frequencies and vortex lattice configurations at matching fields. **a** I–V curves in the vortex velocity versus normalized current representation. The horizontal arrows indicate the vortex velocities at which the emission spectra in Fig. 2 have been acquired. Inset: The I–V curve of the sample at 3 K and 4.2 T. The vertical arrow indicates the definition of $I^*$ used for plotting the I–V curves in the main panel. **b** Peak frequencies versus vortex velocity for the data sets (**c**, **d**) of Fig. 2. **c** Resistance as a function of $H^{\|b}$ for a series of temperatures, as indicated. The vortex lattice configurations, which commensurate with the Mo/Si superlattice at $H_{N=2}^{(2)} = 3.15$ T and $H_{N=1}^{(1)} = 4.2$ T, are shown in the scaled coordinate system ($\gamma a$, c) in the inset. **d** Normalized emission power $P_m$ as a function of temperature for the first four lowest-order harmonics $f_0^{(2)} = 10.02$ GHz emitted at the vortex velocity $v = 100$ m s$^{-1}$ at 3.15 T. Solid lines are guides for the eye

comparison of the emission spectra acquired at different $T$ and $H$ values in Fig. 2 using $v$ as a deduced parameter but also links the peak frequencies with the dc bias current $I$, which is a driving parameter in our experiment. In particular, with increase of the vortex velocity to >20 m s$^{-1}$ a series of peaks appears in all panels of Fig. 2 on the background of the noise floor. The peaks are best seen in the range of vortex velocities between 50 and 150 m s$^{-1}$ corresponding to the nearly linear regime of viscous flux flow in Fig. 3a. We note that the higher-frequency peaks at $f_m = mf_0$ are harmonically related to the lowest-frequency peak at $f_0$, and the detected spectra do not change under dc current polarity reversal, i.e., for entering and exiting vortex rows. The largest number of harmonics $m = 6$ is observed in the accessible frequency range at a vortex velocity $v = 75$ m s$^{-1}$. Except for the data set (a) in Fig. 2, to which we return in what follows, the peak power $P_m$ decreases with increasing $f$. We emphasize that, whereas $f_0$ does not depend on temperature, it does depend on the magnetic field. This is why in what follows we will distinguish between $f_0^{(2)}$ for the data sets (a), (c), and (e) acquired at $H = 3.15$ T and $f_0^{(1)}$ for the data acquired at 4.2 T. Importantly, $f_0^{(1,2)}$ are shifted towards higher frequencies with increase of the dc bias current. Specifically, $f_0^{(2)} = 5.01$ GHz at 3.15 T and $f_0^{(1)} = 9.98$ GHz at 4.2 T at

$v = 50$ m s$^{-1}$ evolve into $f_0^{(2)} = 15.04$ GHz and $f_0^{(1)} = 29.87$ GHz at $v = 150$ m s$^{-1}$, respectively. A linear dependence of the peak frequencies on the vortex velocity becomes apparent in Fig. 3b where the data deduced from panel columns (c) and (d) of Fig. 2 are presented. Evidently, the observed emission is related to the washboard frequency associated with the vortex dynamics.

**Vortex lattice configurations at matching fields.** The $H$ values at which the spectra in Fig. 2 have been acquired correspond to minima in the resistance curves $R(H^{\|b})$ shown in Fig. 3c. Namely, at $T = 3.6$ K, $R(H^{\|b})$ has a minimum centered at $H = 3.15$ T. At $T = 3$ K, the minimum at $H = 3.15$ T becomes deeper and a second minimum appears at $H = 4.2$ T. At $T = 1.8$ K, the two minima evolve into zero-resistance states in fields 3–3.3 and 4.15–4.35 T. For the elucidation of what periodic length scale in the studied system is associated with the peaks at the different $H$, we analyze the stable vortex lattice configurations at the resistance minima in Fig. 3c. Namely, the commensurability effect in anisotropic layered superconductors was considered theoretically by Bulaevskij and Clem (BC)[32] on the basis of the discrete Lawrence–Doniach approach and by Ivlev, Kopnin, and Pokrovsky (IKP) in the framework of the continuous Ginzburg–Landau model[33]. The $R(H^{\|b})$ curve of our sample has

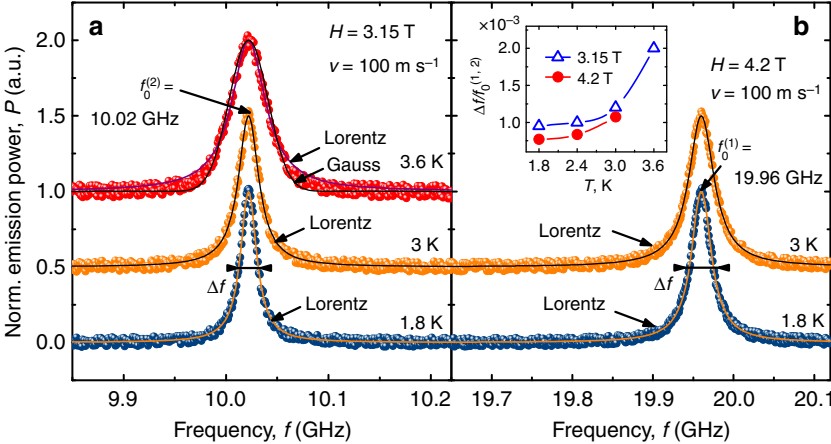

**Fig. 4** Line shape of the emission. Electromagnetic radiation at the washboard frequencies $f_0^{(2)} = 10.02$ GHz at $H = 3.15$ T (**a**) and $f_0^{(1)} = 19.96$ GHz at $H = 4.2$ T (**b**) for a vortex velocity $v$ of 100 m s$^{-1}$ and a series of temperatures, as indicated. Symbols: experimental data; solid lines: Lorentz and Gauss fits. An offset of 0.5 is used along the vertical axis to facilitate reading the data. Inset in **b**: Temperature dependences of the normalized linewidth $\Delta f/f_0^{(1,2)}$ at 3.15 T (triangles) and 4.2 T (circles)

no minima at the BC matching fields, see also Supplementary Note 1. We attribute this to a relatively large interlayer coupling in our sample and compare the data with the continuous IKP model. Namely, the IKP matching fields in our data range are $H_{N=1}^{(1)} = 4.2$ T and $H_{N=2}^{(2)} = 3.15$ T, in perfect agreement with the field values at which the resistance minima are observed in Fig. 3c. We note that so far we have not been able to detect an em emission at smaller field values corresponding to the expected[33] higher-$N$ matching configurations of the vortex lattice. In all, our analysis of the resistance minima, in conjunction with the universal scaling of $I$–$V$ curves in the flux-flow regime, suggests that we deal with a lattice of Abrikosov rather than Josephson vortices. At the same time, we can not rule out a crossover from Abrikosov to Josephson vortices with further decrease of the temperature, as such a crossover is known in layered systems when the Abrikosov vortex with a suppressed order parameter in its core turns into a Josephson phase vortex once its core completely fits into the insulating layer[20]. Namely, in this case the system is expected to transit into the regime of strong layering, with the Josephson vortex lattice configurations described by the BC theory[32]. In addition, such an Abrikosov-to-Josephson vortex transition can be evidenced by a non-monotonic temperature dependence of the flux-flow voltage for the in-plane vortex dynamics in the geometry when **H** ∥ **b** and **I** ∥ **c**[21].

## Discussion

Figure 4 displays the emission peaks at $f_0^{(2)} = 10.02$ GHz at $H = 3.15$ T (a) and $f_0^{(1)} = 19.96$ GHz at $H = 4.2$ T (b) for a vortex velocity $v$ of 100 m s$^{-1}$ and a series of temperatures. In all cases, the peak line shape can be fitted well to a Lorentzian, thus allowing us to introduce the linewidth $\Delta f$. A larger deviation from the Lorentzian is observed at 3.6 and 3.15 T in Fig. 4a, where a Gaussian is added for comparison and the experimental data fall between these two fits. We note that an evolution of the em radiation line shape from Lorentzian to Gaussian is known to occur in Josephson junctions with increase of temperature[34]. Such a Lorentzian-to-Gaussian crossover in the line shape was predicted for Josephson point junctions[35] and flux-flow oscillators[36] when thermal fluctuations broaden the linewidth. We suppose that this might be the case in our system as well. Proceeding to the evolution of the linewidth as a function of temperature and magnetic field value, we note that at

1.8 K $\Delta f^{(2)} = 10$ MHz at $f_0^{(2)} = 10.02$ GHz and $H = 3.15$ T while $\Delta f^{(1)} = 16$ MHz at $f_0^{(1)} = 19.96$ GHz and $H = 4.2$ T. At lower temperatures, the linewidth is almost independent of temperature. By contrast, a notable line broadening occurs at higher temperatures $T \lesssim T_c$. Thus a typical normalized linewidth at $0.5 T_c$ is $\Delta f/f_0^{(1,2)} \simeq 10^{-3}$, i.e., it is by one-to-two orders of magnitude larger than the radiation linewidth from intrinsic Josephson junctions in Bi$_2$Sr$_2$CaCu$_2$O$_8$[37], where record values reach $10^{-5}$[5]. At the same time, $\Delta f/f_0^{(1,2)}$ in our multilayer is by a factor of ten smaller that the linewidth of the em generation from moving Abrikosov vortices in a Nb film picked up by an overlying meander antenna[38]. The normalized linewidth $\Delta f/f_0^{(1,2)}$ at 4.2 T is by about 25% smaller than that at 3.15 T. We attribute this to a more dense vortex lattice at 4.2 T and, hence, a more ordered vortex motion as compared to 3.15 T.

The evolution of the matching minimum in the $R(T)$ curve at $T = 3.6$ K to the zero-resistance state at $T = 1.8$ K in Fig. 3c can be understood with the aid of the superconductivity dimensionality crossover occurring in the Mo/Si superlattice, as inferred from the $H$–$T$ phase diagram shown in Fig. 5a. The out-of-plane upper critical field extrapolated to zero temperature $H_{c2}^{\|c}(0) = 7.4$ T yields $\xi_{ab} = [\Phi_0/2\pi H_{c2}^{\|c}(0)]^{1/2} = 63$ Å and, hence, $\xi_c(0) = \xi_{ab}(0)/\gamma = 12$ Å. In the $H$–$T$ diagram, there is a crossover temperature $T^* = T_c(1-\tau)$ with $\tau = 2\xi_c^2/s^2 \approx 3.60$ K below which the system behaves in a 2D manner and exhibits a three-dimensional behavior at $T > T^*$. The increase of the size of the vortex core with increasing temperature $\simeq 2\xi_c(T)$ is illustrated in Fig. 5b in comparison with the thickness of the Si layer $d_{Si}$ and the multilayer period $s$. As a brief summary of Supplementary Note 2 devoted to the superconductivity dimensionality crossover with increasing temperature, a semi-quantitative relation of the vortex core size to the Si layer thickness and the multilayer period is sketched on the top of the spectra in Fig. 2. In particular, at 1.8 K, being the lowest temperature accessible in our experiment, the vortex core $2\xi_c(1.8K) \approx d_{Si} = 28$ Å largely fits into the insulating layers, thereby allowing the Mo layers to remain superconducting up to very high fields[39,40]. At 3 K, the vortex core $2\xi_c(3 K) \approx s \approx 50$ Å becomes comparable with the multilayer period. Even though some part of the vortices penetrates into the Mo layers, there are field ranges where the intrinsic pinning energy $E_p$ is larger than the elastic energy of a vortex lattice shear deformation $E_{el}$, which explains the presence of a rather broad resistance minimum in

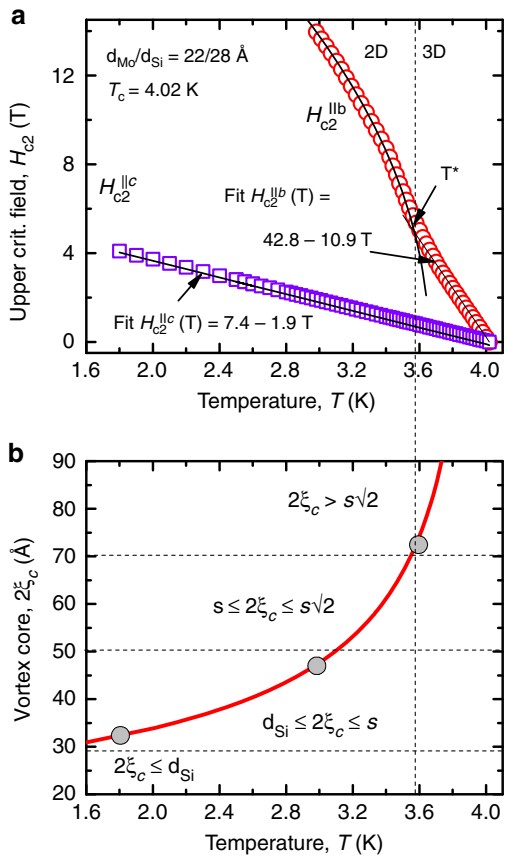

**Fig. 5** Superconductivity dimensionality crossover in the Mo/Si superlattice. **a** The in-plane $H_{c2}^{\|b}$ and out-of-plane $H_{c2}^{\|c}$ upper critical fields versus temperature. Solid lines are fits $\propto (T_c - T)$ in the 3D regime and $\propto (T_c - T)^{1/2}$ in the 2D regime. The 2D–3D crossover temperature $T^*$ corresponding to $\xi_c(T^*) = s/\sqrt{2}$ is indicated. **b** Temperature dependence of the vortex core size $\simeq 2\xi_c$ with the different regimes determined by the relation of $\xi_c$ and the multilayer period $s$. Large circles indicate the temperature at which the emission spectra in Fig. 2 have been acquired

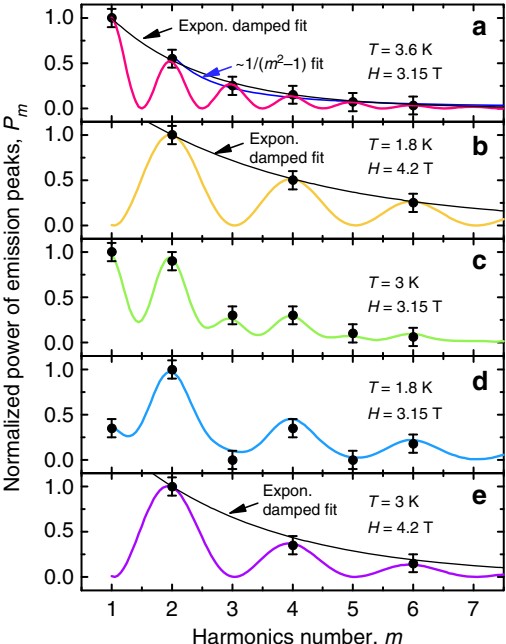

**Fig. 6** Intensity of higher harmonics in the microwave radiation. Normalized emission power $P_m$ as a function of the harmonics number $m$ for a vortex velocity of $v = 75\,\mathrm{m\,s^{-1}}$. The assumed superposition of the em wave emission associated with the 100 Å spacing between the vortex rows (**a**) and the 50 Å-periodic layered structure (**b**) is used to explain the spectrum modification in **c**, **d**. The spectrum at 3 K (**e**) exhibits a stronger damping of higher harmonics as compared to 1.8 K (**b**). Symbols are $P_m$ values deduced from the spectra in Fig. 2. Solid lines are fits as indicated. Error bars depict the doubled standard deviation encompassing about 95% of the data

the vicinity of the matching fields. At 3.6 K, the vortex cores become appreciably larger than the multilayer period, namely, $2\xi_c(3.6\,\mathrm{K}) > 70$ Å, such that the intrinsic confinement potential is smoothed out as the vortex core extends over more than one multilayer period. In this case, the superlattice is no longer felt by a vortex as a layered structure but rather the motion of vortices occurs in some effective continuous medium. Accordingly, the matching minimum at 3.15 T becomes shallow at 3.6 K while the minimum at 4.2 T disappears altogether as this field value is too close to $H_{c2}^{\|b}(3.6\,\mathrm{K}) = 5.2$ T and it gets smeared by the transition to the normal state.

The relation of the vortex core size $\sim 2\xi(T)$ to $d_{Si}$ and $s$, as discussed above, allows for the following explanation of the differences in the microwave emission spectra in Fig. 2, where the major experimental findings are (i) the frequency doubling when changing the field value from 3.15 to 4.2 T and (ii) the evolution of the amplitude-versus-harmonics-number enveloping function from the exponentially damped curve at 3.6 K to a non-monotonic curve at 1.8 K for the data set at 3.15 T, see Fig. 6. The relation of the peak frequency to the vortex velocity $f_0^{(1,2)} = v/d^{(1,2)}$ suggests that the washboard period $d^{(2)} = 100$ Å is involved in all spectra at 3.15 T while another washboard period $d^{(1)} = 50$ Å is involved in all spectra at 4.2 T. In addition, Fig. 6c, d resemble interference patterns (to be discussed in what follows), that makes us to assume that emissions

associated with both periods $d^{(1)}$ and $d^{(2)}$ are involved at 3.15 T at 1.8 and 3 K.

In general, possible em emission sources in our sample are (i) its front surface, (ii) its back surface, and (iii) intrinsic superconducting/insulating interfaces. We note that the possibility for an em emission from internal superconducting/insulating interfaces goes significantly beyond the theory developed for the case of a homogenous superconductor[28]. In addition, emission sources (ii) and (iii) bring up the question of propagation of em waves through the Mo/Si multilayer. In particular, using the relation $\lambda(0) = 1.05 \times 10^{-3} \rho_0^{1/2} T_c^{-1/2}$ given by Eq. (A13) from ref. [41], where $\rho_0$ is the resistivity just above $T_c$, we come with 740 nm as an estimate for the penetration depth at zero temperature in our sample. For the in-plane field component, this yields $\lambda_{ab}(0) = 325$ nm with $\lambda_{ab}(1.8\,\mathrm{K}) = 900$ nm, $\lambda_{ab}(3\,\mathrm{K}) = 1.25\,\mu\mathrm{m}$, and $\lambda_{ab}(3.6\,\mathrm{K}) = 1.9\,\mu\mathrm{m}$, i.e., the penetration depth is by a factor of 3–8 larger than the multilayer thickness amounting to 250 nm. This means that, in principle, em waves can propagate through the entire sample and we can not exclude an interference of the radiation from the front sample surface with that coming from its back surface as well as from the intrinsic superconducting/insulating layers. Unfortunately, the existing theories[27,28] do not make predictions for the contributions from sources (ii) and (iii) in our experimental system. At the same time, on the basis of our experimental data we can not judge whether and how large contributions to the recorded spectra do sources (ii) and (iii) provide. A theory including the appropriate boundary and periodic conditions of whether and how this radiation arises and propagates through the multilayer is yet to be developed. At the same time, a qualitative explanation of the presence of the em emission associated with a washboard period of 50 Å at 3.15 T

and 1.8 and 3 K can be suggested on the basis of the intrinsic 50 Å periodicity of the multilayer itself. Namely, when the vortex rows cross the sample surface with the washboard period $d^{(2)} = a^{(2)}\sqrt{3}/2$, the redistribution of supercurrents between the layers occupied by vortices and vortex-free layers is characterized by the halved washboard period $d^{(2)}/2 = s$. Hence, the redistribution of supercurrents with a doubled frequency coexists with the surface-crossing frequency at 3.15 T at lower temperatures. By contrast, when the vortex cores become larger than the multilayer period, the vanish of the doubled frequency contribution at 3.6 K and 3.15 T can be explained by smearing of the current lines, which flow into adjacent layers as the vortex core increases and is moving in a smoothed out periodic pinning potential.

In this way, at 3.6 K our experiment is most closely related to the problem considered theoretically[28], and in this case we observe an emission of em waves at the harmonics of the washboard frequency $f_m^{(2)} = m f_0^{(2)} = mv/d^{(2)}$ with $d^{(2)} = 102$ Å nicely corresponding to the matching condition $2s = d^{(2)} = a^{(2)}\sqrt{3}/2$ for the triangular flux lattice with the parameter $a^{(2)} = \left(2\Phi_0/\sqrt{3}H_{N=2}^{(2)}\gamma\right)^{1/2} \approx 115$ Å at $H_{N=2}^{(2)} = 3.15$ T, Fig. 2e. By contrast, in the case of vortices whose diameters are smaller than the multilayer period, this 100 Å-period-related em emission is superimposed with the 50 Å-period-related emission. Furthermore, the data in Fig. 2b, d corroborate that the emission of em waves becomes also possible at harmonics of $f_m^{(1)} = m f_0^{(1)} = mv/d^{(1)}$ with $d^{(1)} = 49.2$ Å corresponding to $a^{(1)} = \left(2\Phi_0/\sqrt{3}H_{N=1}^{(1)}\gamma\right)^{1/2} \approx 100$ Å with the matching condition $a^{(1)} = 2d^{(1)} = 2s$ at $H_{N=1}^{(1)} = 4.2$ T. At the same time, the em emission peaked at halved frequency $f_m^{(2)} = m f_0^{(2)}$ and, hence, related to a doubled periodic length scale as compared to $f_0^{(1)}$ is clearly distinguishable at $H_{N=2}^{(2)} = 3.15$ T. Finally, when the vortex cores become larger than $d_{Si}$ (and especially larger than $s$) the softening of the spatial profile of the order parameter results in that the emission associated with the 100 Å washboard period starts to dominate the emission related to the 50 Å multilayer periodicity, Fig. 2e.

To support the assumption that in the general case the emission can be presented as a superposition of 100 Å- and 50 Å-period-related emissions, in Fig. 2 we denote the odd harmonics of $f_0^{(1)}$ with $m^{(1)} = 1, 3, 5$ by solid triangles, the even ones with $m^{(1)} = 2, 4, 6$ by open triangles, and the odd and even harmonics of $f_0^{(2)}$ with solid and open circles, respectively. The normalized power $P_m$ of the emitted harmonics as a function of the harmonics number $m$ is plotted in Fig. 6. We note that $P_m(m)$ follows an exponential decay for the assumed dominating emission related to the 100 Å-periodic vortex row spacing in Fig. 6a as well as for the assumed dominating emission related to the 50 Å-periodic superlattice in Fig. 6b, e. By contrast, the patterns of $P_m(m)$ at 3.15 T at lower temperatures are non-monotonic, Fig. 6c, d. We note that a decay of higher harmonics following the law ~$1/(m^2 - 1)$, which is a very good approximation to the exponentially damped fit for $m \leq 2$, was observed for the harmonics generation upon microwave transmission through granular YBCO thin films in the presence of an alternating current (ac) magnetic field[42]. Interestingly, an attenuation of higher harmonics in the electric field response in superconductors with a washboard pinning potential in the presence of a combination of dc and ac currents has been predicted to follow the envelope of modified Bessel functions[43]. Owing to the mathematical analogy of the Langevin equation of motion of an Abrikosov vortex with the equation for the phase change in a Josephson contact, the microwave Shapiro step amplitude follows the same law as a function of the microwave voltage in small junctions[44].

If we now treat the exponentially damped curves in Fig. 6a, b as functions enveloping harmonic functions with the period $p = m$ for the emission related to the 100 Å-periodic length scale in (a) and $p = 2m$ for the 50 Å-periodic length scale in (b), the $P_m$ pattern in panel (c) fits to a superposition of 0.32/0.68-weighted functions from panels (a) and (b), while a good fit for $P_m$ in panel (d) is found for a superposition of 0.69/0.31-weighted functions from panels (a) and (b), available in the Source Data. This allows for treating the $P_m(m)$ dependences in panels (c) and (d) as beatings of the $f_0$- and $2f_0$-waves, which suggests that the 50 and 100 Å periods produce out-of-phase contributions in the resulting spectrum. In particular, the assumed out-of-phase contributions of the $f_0$- and $2f_0$-waves allows us to explain the absence of peaks at $m = 3, 5$ at $T = 1.8$ K in the data set (a) in Fig. 2. The gradual decrease of the odd harmonics of $f_0^{(2)}$ accompanied by a simultaneous growth of the even ones at decreasing temperature becomes apparent in Fig. 3d. Thus a suitable choice of the temperature allows for a frequency-selective generation of em waves evolving from the high-temperature spectrum containing a series of exponentially damped higher harmonics over an intermediate-temperature spectrum with higher harmonics obeying a more complex, beating-related law to the low-temperature spectrum in which the higher-order odd harmonics are absent. Furthermore, a faster attenuation of $P_m$ at 3 K in Fig. 6d as compared to 1.8 K in Fig. 6b can be attributed to softening of the spatial profile of the order parameter whose Fourier transform contains a smaller number of higher harmonics. Finally, the strongly suppressed em radiation at vortex velocities <5 m s$^{-1}$ might indicate that the typical time $t = 1/f = \Delta d/v \gtrsim 1 \times 10^{-11}$ s of restoring the superconducting condensate upon crossing the edge ($\Delta d \sim 5$ Å) of a superconducting layer by vortices becomes sufficiently larger than the quasiparticle relaxation time in the studied system. This means that the variation of the magnetic induction as the vortices leave and enter the superconducting layers occurs adiabatically that can explain the absence of an emission in this quasistatic regime.

The dc-to-microwave power conversion efficiency in the investigated system is at present very low. Indeed, if we introduce the conversion efficiency parameter $\kappa = P^{mw}/P^{dc}$, where $P^{dc}$ and $P^{mw}$ are the supplied and detected microwave powers, respectively, we obtain $\kappa \simeq 10^{-7}$ for $v = 100$ m s$^{-1}$ at 3.15 T and 3 K. For comparison, THz generators based on single Josephson junctions have $\kappa \simeq 10^{-5}$–$10^{-3}$, typically, while for stacks of Josephson junctions coupled to a resonator the conversion efficiency can reach a few tens percent[5,13,45]. Accordingly, approaches to improve the conversion efficiency in the investigated system should be developed. For a single- or discrete-frequency generation, coupling of the system to a resonator should be considered. For operation in the tunable-frequency mode, efforts might be directed to the formation of a square vortex lattice, which is expected to be a more efficient em emitter as compared to a (squeezed) triangular vortex lattice[46].

As an implication for superconducting applications, which can be drawn from our study, superconductor/insulator multilayers posses a potential for the use as on-chip generators. Their emission frequency $f_m = m f_0 = mv/d$ with $d = s$ (or $d = 2s$, depending on the magnetic field value) can be monitored via the voltage drop related to the vortex velocity $v$ and be continuously tuned by the transport current, which is a driving parameter, via the relation $v = 109.5(I/I^* - 1)$ m s$^{-1}$. The in-plane layout of Mo/Si superlattices allows for their on-chip integration with other fluxonic devices, such as diodes[47], microwave filters[48], and transistors[49] operating with Abrikosov vortices as well as quantum devices exploiting Josephson vortices as building blocks for coherent terahertz generation[5] and qubits for quantum computing[50].

To summarize, we have observed microwave radiation from a lattice of Abrikosov vortices moving across the layers in a Mo/Si superlattice. The emission spectrum is peaked at the harmonics of the washboard frequency $f_0^{(1)}$ related to the multilayer period and $f_0^{(2)}$ associated with the distance between the vortex rows in the direction of motion. The emission spectrum can be finely tuned by the dc bias current and, coarsely, by switching the in-plane magnetic field between matching values. In addition, we have revealed that the emission spectrum evolves as a function of temperature, such that the odd harmonics of the washboard frequency related to the distance between the vortex rows can be almost completely suppressed by choosing the matching field at which the vortex lattice is pinned in all neighboring insulating layers at lower temperatures. In all, our findings suggest that superconductor/insulator superlattices can act as dc-tunable microwave generators bridging the frequency gap between conventional rf oscillators and (sub-)terahertz generators relying upon the Josephson effect.

## Methods

**Fabrication and properties of the Mo/Si superlattice.** The superconductor/ insulator superlattice consists of 50 Mo and Si bilayers alternately sputtered onto a glass substrate at a substrate temperature of 100 °C. The deposition rate was 2 Å s⁻¹. The thicknesses of the amorphous Mo and Si layers are $d_{Mo} = 22$ Å and $d_{Si} = 28$ Å, resulting in a superconducting layer repeat distance $s$ of 50 Å, which is referred to as a multilayer period. The individual layer thicknesses were inferred from small-angle X-ray reflectivity with an accuracy 0.1 Å. The sample has 10 nm-thick top and bottom Si layers. Its superconducting transition temperature, determined at the midpoint of the resistive transition $R(T)$, is $T_c = 4.02$ K. This is noticeably higher than $T_c = 0.92$ K of bulk Mo because of oscillations of $T_c$ of Mo/Si multilayers with increasing $d_{Si}$ and an eventual saturation at $T_c = 7$ K for $d_{Si} > 120$ Å[51]. The interlayer Josephson coupling in the Mo/Si superlattice studied here is rather strong $\eta_J = \hbar^2/2ms^2\gamma^2 \approx 1$[15]. A large ratio of the effective mass of the Cooper pairs $M$ perpendicular to the layer planes to the in-plane mass $m$ gives rise to an anisotropy $\gamma = (M/m)^{1/2} \approx 5.72$ of the physical parameters of the superlattice, as inferred from the $H–T$ phase diagram shown in Fig. 5a and detailed in Supplementary Note 2. These parameters include the in-plane ($ab$) and out-of-plane ($c$) upper critical field $H_{c2}^{||ab} = \gamma H_{c2}^{||c}$, the coherence length $\xi_{ab} = \gamma\xi_c$, and the penetration depth $\lambda_c = \gamma\lambda_{ab}$. The structure of individual vortices and the vortex lattice in the sample differs in essential ways from the conventional triangular vortex lattice in homogenous isotropic superconductors[20]. Namely, the vortex core is elongated in the layer planes and compressed along the $c$-axis. For a magnetic field applied along the $b$-axis, the ground state vortex lattice configuration is given by a regular triangular lattice in the scaled coordinates ($a\gamma$, $c$)[33,52], which are used in all sketches in Fig. 2 and in the inset of Fig. 3c. A low-bound estimate for the zero-temperature gap frequency $2\Delta_0/h$ of the studied Mo/Si sample can be done using the standard BCS weak-coupling relation $\Delta_0 = 1.76k_BT_c$, which yields $f_G(0) \simeq 300$ GHz and $f_G \simeq 100$ GHz at 3.6 K, i.e., well above the highest frequency accessible in our experiment.

**Fabrication of contacts.** Care has been taken to ensure a homogenous current distribution over individual layers in the Mo/Si sample. Namely, individual layers were interconnected in the location of each of four current and voltage leads in accordance with the fabrication steps illustrated in Supplementary Fig. 1. The interconnection was done by a combination of focused ion beam induced deposition (FIBID)[53] and focused Ga ion beam (FIB) milling[54] in a high-resolution dual-beam scanning electron microscope (SEM: FEI, Nova Nanolab 600). First, a protective Pt-FIBID layer was deposited on top of the Mo/Si multilayer, Supplementary Fig. 1a, to avoid edge rounding and poisoning of the top layers of the Mo/Si superlattice by the Ga ions. The dimensions of the protective Pt-FIBID layer were 50 μm × 5 μm × 500 nm (length×width×height). The deposition was done employing the precursor gas (CH₃)₃CH₃C₅H₄Pt at 30 kV/100 pA beam parameters. Next, a stair-like groove was milled by FIB in the Mo/Si multilayer under normal beam incidence and then the sample was tilted by 52°, Supplementary Fig. 1b, to remove a thin layer of redeposited Mo–Si–Ga on the side wall of the milled groove, Supplementary Fig. 1c. The dimensions of the milled region were 50 × 5 × 2 μm³ and the beam parameters were 30 kV/1 nA. The "cleaning cross-section" had dimensions 55 μm × 2 μm × 2 μm and was done at 30 kV/300 pA. Finally, a conducting Pt-FIBID layer was deposited on the side wall (50 μm × 1.5 μm × 500 nm, 30 kV/50 pA), covering the area from the top protective layer down to the substrate. Exemplary SEM images of the sample surface after two last preparation stages are shown in Supplementary Fig. 2. The same nanofabrication techniques were used for the lamella preparation for a transmission electron microscopic inspection.

**Ultra-wide-band cryogenic spectroscopy.** A four-probe 5 × 1 mm² bridge was patterned in the sample for electrical transport measurements. The distance between voltage contacts amounted to 2.44 mm. The magnetic field and transport current were applied in the layer plane and orthogonal to each other, causing a vortex motion across the layers under the action of the Lorentz force. The measurements were performed in a ⁴He cryostat with a magnetic field provided by a superconducting solenoid and directed parallel to the layer planes. The magnetic field misalignment error was 0.2°. Owing to the lock-in transition, as outlined in Supplementary Note 3, vortices run strictly parallel to the layers and emission is independent of the field misalignment angles between −1° and +1°. The dc voltage and the emitted microwave power were measured simultaneously by a nanovolt-meter and a spectrum analyzer in the frequency range from 100 MHz to 50 GHz. The microwave spectrometer allowed for the detection of signals with power levels down to $10^{-16}$ W in a 25 MHz bandwidth. The spectrometer system consisted of a spectrum analyzer (Keysight Technologies N9020B, 10 Hz–50 GHz), a semirigid coaxial cable (SS304/BeCu, dc–61 GHz, insertion loss 6.94 dB/m at 20 GHz), and an ultra-wide-band low-noise amplifier (RF-Lambda RLNA00M54GA, 0.01–54 GHz). The emitted signal was picked up by a wire loop shorting the end of a semirigid coaxial cable and placed close to the sample edge parallel to the sample surface. The diameter of the wire loop was about 2.8 mm such that the antenna operated in a nongradient loop coupling[31] in the whole accessible frequency range. The signal was amplified by a low-noise preamplifier with a gain of 36 dB. The signals emitted at different temperatures were further normalized by attenuators. The attenuation levels were set to −15 dB at $T = 1.8$ K, −12 dB at 3.0 K, and −1.5 dB at 3.6 K. The frequency dependence of the coupling strength of the sample-antenna-transmission-line system was corrected using the flux-flow oscillator itself. Namely, the amplitude of the peak at the first harmonics at given $T$ and $H$ values was recorded as a function of the dc current in the range of vortex velocities between 20 and 250 m s⁻¹, corresponding to the regime of viscous flux-flow and to the linear section of the $I–V$ curve. Unevennesses in the recorded frequency dependence in this regime are related to spurious resonances/reflections in the sample-antenna-transmission-line system. The frequency dependence of the detected amplitude thus recorded was saved in the analyzer as a reference floor for the amplitude-sensitive representation of emission peaks at higher harmonics. The frequency dependence of the picked up amplitude has been revealed to depend very weakly on temperature and magnetic field in their investigated ranges. This has allowed us to correct for the frequency dependence of the coupling strength of the pick-up loop to the sample in the frequency range between 5 and 50 GHz.

## Data availability

The data sets generated and/or analyzed during the current study are available from the corresponding author on reasonable request.

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

## Acknowledgements

O.V.D. thanks Vasyl Denysenkov for useful advices at preliminary stages of the experiment and acknowledges the German Research Foundation (DFG) for support through Grant No 374052683 (DO1511/3-1). This work was supported by the European Cooperation in Science and Technology via COST Action CA16218 (NANOCOHYBRI). Also funding from the European Commission in the framework of the program Marie Sklodowska-Curie Actions—Research and Innovation Staff Exchange (MSCA-RISE) under Grant Agreement Nos. 644348 (MagIC) and 645660 (TUMOCS) is acknowledged.

## Author contributions

O.V.D. and V.A.S. conceived the experiment. M.Y.M. and O.I.Y. fabricated the samples and assisted with selected measurements. O.V.D. performed the measurements with assistance from M.I.T. and R.S. V.M.B., R.V.V. and M.H. evaluated the emission spectra. All authors discussed the results and commented on the manuscript at all stages. O.V.D. wrote the manuscript with inputs from all authors.

## Additional information

**Competing interests:** The authors declare no competing interests.

