## [Peer Review File · Nature Communications]

Reviewers' comments:

Reviewer #1 (Remarks to the Author):

see attached report

Reviewer #2 (Remarks to the Author):

See attachment

Reviewer #3 (Remarks to the Author):

The manuscript by Dobrovolskiy et al. describes the generation of electromagnetic waves in the gigahertz frequency band by driving superconducting vortices in Mo/Si superlattices. The authors show that superconductor/insulator superlattices can act as dc-tunable microwave generators.

The superconducting gap in high T_c cuprate superconductors is of several meV which corresponds to several terahertz (THz) in the electromagnetic (EM) frequency band. It is widely believed that the superconductors can be used to generate continuous THz EM waves, which has wide range of applications. There are tremendous efforts in the past two decades, aiming at stimulating strong terahertz emission. The THz EM waves have been successfully observed by experiment in 2007, which was comprehensively reviewed in Ref. 12 of the manuscript. The radiation was detected in BSCCO superconductor through the intrinsic Josephson effect. Meanwhile an interesting theory on the generation EM wave by driving superconducting vortices in type II superconductors has been proposed by Bulaevskii and Chudnovsky in 2006. Now this proposal is eventually realized by the experiments presented in the manuscript.

Instead of using a bulk superconductor as proposed in the theory, Dobrovolskiy et al. fabricated Mo/Si superlattices. By applying magnetic field in the b axis of the superconducting superlattice, vortex lattice is induced. The authors have identified the resulting vortex lattice is of Abrikosov vortex lattice by identifying the resistance minima. They then applied a dc current in the a axis to drive the vortex lattice in the c direction. The vortex lattice experiences the washboard potential at the superlattice edge, and also the periodic potential produced by the superlattice itself. The motion of vortex lattice in both potential generates EM wave but with distinct frequencies. The authors have observed both radiation. The dependence of the radiation frequency on magnetic field and temperature is discussed and is consistent with what one would expect for the radiation generated by the vortex motion. Thus the present work establishes firmly as the first experimental detection of EM wave radiation due to the motion of the Abrikosov lattice. Note that the EM emission due to the Josephson vortex flow was detected in BSCCO in Bae et al. Phys. Rev. Lett. 98, 027002 (2007).

I congratulate the authors for the accomplishment, and would like to recommend strongly the manuscript for publication. The manuscript can be published as it is.

Finally some minor technical remarks:

- 1) In p2, the unit of magnetic flux should be Wb.
- 2) In p3, the symbols in η_J are not defined, although they are later defined in the discussion section. It is better to define them right after η_J .
- 3) (Optional) One advantage to generate EM waves using the vortex lattice is that the frequency can be tuned continuously by magnetic field and current. However, the radiation power is weak (of the

order of picowatt). This is probably due to the triangular lattice of Abrikosov vortex, where the radiation from different parts of the system is out-of-phase and interfere destructively. It is helpful to discuss how to enhance the radiation by achieving a rectangular arrangement of vortices.

We would like to thank all referees for their interest in our work and their comments. In the revised version of the manuscript, the changes made are highlighted in blue.

Reviewer #1 (Remarks to the Author):

1. Point:

In order to account for features in the observed spectra, the authors introduce the notion of em-emission from internal sc-insulating interfaces. This description goes beyond the model described in Ref. 9, and they should add a detailed description of how this radiation arises, including the appropriate boundary conditions. Following Fig. 1 in Bulaevskii & Clem PRB 44, 1023 (1991) (BC), on first glance one would say that, averaged over the scale of the em-wavelength, oscillating currents largely cancel, and that there is no emission from internal interfaces. They should also outline how this radiation would propagate through the sc-multilayer. If there were indeed interference with radiation from internal interfaces, is there also interference with radiation coming from the back surface of the sample? What is the penetration depth of the sample?

Answer:

We agree with the point of the referee. The interpretation of the emission spectra relying upon the notion of em emission from internal sc-insulating interfaces goes significantly beyond the BC model. A theory for em emission from internal sc-insulating interfaces is not available, so that we reconsidered our interpretation of the interference and modified the discussion section accordingly.

Just to recall, what are most relevant experimental findings in Fig. 2 in this regard? This are

- frequency doubling when changing the field value from 3.15 T to 4.2 T and
- evolution of the amplitude-versus-harmonics-number enveloping function from the exponentially-damped curve at 3.6 K to a non-monotonic curve at 1.8 K for the data set at 3.15 T.

The relation of the peak frequency to the vortex velocity suggests that the 10 nm washboard period is involved in all spectra at 3.15 T and another washboard period of 5 nm is involved in all spectra at 4.2 T. In addition, Fig. 5(c) and (d) are reminiscent of interference patterns. Therefore, we *assume* that emissions associated with *both* periods are involved at 3.15 T at 1.8 K and 3 K.

Now we ask ourselves: Radiation from what sources is available in the investigated system? In general, possible sources of radiation in our experiment are:

(i) the front surface, (ii) the back surface, and (iii) intrinsic sc/insulating interfaces

Using Eq. (A13) from Kes and Tsuei, PRB 28, 5126 (1983) we estimate the magnetic penetration depth at zero temperature 740 nm resulting in $\lambda(0) = 325$ nm in the direction of vortex motion, with $\lambda(1.8 \text{ K}) = 900$ nm, $\lambda(3 \text{ K}) = 1.25 \mu\text{m}$, and $\lambda(3.6 \text{ K}) = 1.9 \mu\text{m}$, i.e. λ is by a factor of 3 to 8 larger than the total thickness of all layers, which is 250 nm. This means that in principle, em radiation can propagate through the entire sample and we can not exclude an interference with radiation coming from the back surface of the sample as well as with that coming from the intrinsic sc-insulating layers if it exists.

On the basis of our data we can not conclude how large contributions to the recorded spectra do both sources provide. To resolve this issue, a theoretical account is required and we appeal to theorists to address this question. As for Fig. 1 in PRB 44, 1023 (1991), this sketch is in neglect of distortions of current lines at the sample edges and, in fact, in the course of a discussion with all co-authors we have ended up with a more simple interpretation of the observed interference, which is based on the 5-nm period of the multilayer itself, as is exemplified in the neighboring Fig. 2 on the same page of PRB 44: Namely, even though the BC work is for the case of weak interlayer coupling, we can still use Fig. 2 for a qualitative interpretation of the interference in Fig. 2(a) and 5(d) in our work. In particular, we note that while crossing of the sample surface by vortices occurs with the washboard period $l/2$ (in the BC notation in Fig. 2), the redistribution of currents (long arrows around vortices are interchanged with short arrows in layers without vortices once a row of vortices is exiting the sample) takes place with the *halved washboard period* $l/4$

(*doubled washboard frequency*). Indeed, this allows us to explain the observed interference in our work *without the notion of emission from intrinsic sc-insulating layers*. Namely, the redistribution of supercurrents with a doubled frequency at the sample surface due to the intrinsic 5nm-periodicity of the multilayer itself is a sufficient condition for the presence of emission at the doubled frequency at lower temperatures at 3.15 T. Furthermore, the vanish of the doubled- f_0 contribution with increase of temperature in the spectrum at 3.6 K is consistent with smearing of the current lines which flow into adjacent layers as the vortex core increases.

Changes made:

We have modified the "Modification..." section on the basis of the arguments outlined above.

2. Point:

The authors present emission spectra in Fig. 2 and a detailed description of possible vortex configurations, largely along the lines of their earlier work in Ref. 3 and 4 of the Supplement. However, there is no information given on the characteristics of the radiation such as line width and line shape and their field and temperature dependence, polarization state of the radiation. This information should be added to the manuscript.

Answer and changes made:

At present we cannot rotate our antenna technically. Therefore, we have addressed all the referee's points with exception of the polarization state. Namely, we have added a text paragraph and new Fig. 4 devoted to the line shape and linewidth of the radiation, as well as their dependence on temperature and magnetic field in Discussion.

3. Point:

Are the spectra the same for exiting and for entering vortex rows, that is, for positive and for negative dc-currents ?

Answer:

Yes, they are. We have not revealed any changes in spectra under dc current polarity reversal.

Changes made:

We have added this sentence in the section "Microwave emission...".

4. Point:

Why is the value of d in Figs. 2b) and d) chosen as $d = a = 2s$. Considering the washboard mechanism, one would have thought that d is half this value for the depicted vortex configuration.

Answer:

The referee is absolutely right. We thank the referee for bringing to our attention this unfortunate misprint. While the calculations have not been affected by this error in the sketch, we have checked for and corrected further misprints appeared in the course of typesetting the manuscript.

Changes made:

- in Fig. 2, the sketch in (b) has been corrected.
- in the label of Fig. 2, the factor "2" has been added to "d" for the configuration at 4.2 T.
 - in the paragraph "Modification of the emission spectra...", the anisotropy factor " γ " has been added in all formulae for the vortex lattice parameter (where it was missing) and the typo in the matching condition " $2s = d$ " at 4.2 T has been corrected for $2s = 2d$.

5. Point:

Judging from the geometry of their sample, in the ideal (theoretical) situation the magnetic field should be aligned with the sample to 10^{-6} , which is not practical, implying that the vortex lattice has kinks and jogs. The authors should give a description of their sample alignment, how they determine the alignment, and present angular dependent data of the emission.

Answer:

Fortunately, for the parallel field geometry in layered superconductors there is a *lock-in transition* which lifts this not practical alignment restriction. We refer to Feinberg and Villard, PRL 65, 919 (1990) (FB) and Tinkham *Introduction to Superconductivity* Chapter 9.3.2, McGraw Hill Book Co., NY, 2nd edition. In the FB work it was shown that in the general case when H is applied at a finite angle θ out of planes, there is a finite lock-in angle θ_c , such that when $\theta < \theta_c$ the flux lines run strictly parallel to the planes, remaining locked in between the layers. At $T = 0.9 T_c$, $\theta_c \approx 1^\circ$ and it increases to $\theta_c \approx 2^\circ - 3^\circ$ with decrease of temperature in our sample, Yuzepovich et al., Low Temp. Phys. 26, 103 (2000). A fingerprint of that the field tilt angle θ is smaller than θ_c is the positions of the R minima which do not shift with increase of θ , Mikhailov et al. Low Temp. Phys. 31, 244 (2005). And this is the case in our measurements. In addition, in experiment we are able to adjust α with a misalignment error of $\pm 0.1^\circ$. Due to the lock-in effect, emission is independent of θ for angles between -1° and $+1^\circ$. We are not going to discuss the disappearance of the emission at larger angles $1^\circ < \theta < 90^\circ$ in this paper, since such a high-frequency experiment implies a different sample housing and antenna.

Changes made:

We have added a comment on the lock-in transition in the system, stated the alignment error in our setup, and mentioned that there is no angular dependence of the emission on the field tilt angle at such small misalignments.

6. Point:

There seems to be a sign-error in the equation describing the data in Fig. 3a.

Answer:

The sign is correct. The fit is in fact $\sim 1/(n^2-1)$ with a pre-factor of 1.8 rather than simply $1/(n^2-1)$.

Changes made:

The label has been corrected. In the text $1/(n^2-1)$ has been corrected for $\sim 1/(n^2-1)$.

7. Point:

On page 9, the text refers to 1.8K-data in Figs. 2b and 2d. Fig. 2d shows 3K-data.

Answer:

The referee is right. We thank the referee.

Changes made:

The words "at T = 1.8 K" have been removed.

8. Point:

The authors attempt to relate their discovery to potential applications. In this regard, it might be preferable to give a context of em-emission from superconducting devices in which the high-performance, deployable vortex flow emitters by Koshelets et al. should be mentioned, rather than the lengthy recount of the history of vortices in superconductors in the beginning.

Answer and changes made:

We have rewritten the first paragraph of Introduction in accordance with the referee's recommendation.

Reviewer #2 (Remarks to the Author):

9. Point:

Although it is surely low, what are some typical conversion efficiencies from d.c. power supplied and the (inferred) microwave energy delivered?

Answer:

If we introduce the conversion efficiency coefficient $\kappa = P^{\text{mw}}/P^{\text{dc}}$, where P^{mw} and P^{dc} are the detected and supplied powers, we can estimate $\kappa \sim 10^{-10} \text{ W}/10^{-3} \text{ W} = 10^{-7}$ (we took $v = 100 \text{ m/s}$ and $H = 3.15 \text{ T}$ at 3 K), that is indeed very low. This is much smaller than the typical efficiency of generators based on single Josephson junctions where it is $\sim 10^{-5}$ to 10^{-3} and stacks of Josephson junctions coupled to a resonator, where it can achieve 30%.

Changes made: In the last paragraph before Conclusions, we have added this estimate and a couple of sentences on how the conversion efficiency can be improved.

10. Point:

The authors study two different commensurate flux pinning states. Other possible states are discussed in the Supplementary Section but apparently resistance minima were not seen (a field limitation is apparently blocking their seeing some of these). But one could none the less look for emission at those fields. (They might be frozen in by cooling through T_c in the appropriate fields.) Was this tried?

Answer: Yes, this was tried, but we would like to publish this elsewhere since this was done using another, modulated-detection scheme. At smaller fields there are matching configurations that are described by a non-integer matching index $n = 2.5$. so that there is a mixture of emissions in the spectra associated with different washboard periods. The emission power at this configuration is on the order of 10^{-16} to 10^{-14} W .

Changes made: We have added a sentence that we have not been able to detect emission at high-order matching fields (with the detection scheme used in the paper).

11. Point:

The parameter γ (the effective mass ratio) is introduced early on whereas it defined only later on (and presumably determined from critical field slopes).

Changes made: The parameter γ has been defined immediately after its first appearance.

12. Point:

It is not clear how the current leads are attached.

Changes made:

We have added a detailed description of the fabrication of contacts in the Supplementary Materials and have briefly outlined how the leads were attached in the Methods section.

13. Point: There is a discussion of a possible transition between Abrikosov and Josephson lattices. Is this just a qualitative distinction or is there some discontinuous change? A sentence or two clarifying what they mean here would be useful. Are they referring to where the cores are located or just how big they are. In this connection they should identify in the caption of Fig. 2 which material the "blue" layers are.

Answer: We mean that when the diameter of the vortex cores becomes smaller than the Si layer thickness at $T < 1.8 \text{ K}$, the system transits into the regime of strong layering, Bulaevskii & Clem, PRB 44, 10234 (1991). In this regime, matching minima at 5.27 T and 1.975 T are expected for Josephson vortices (core-less, phase vortices). We do not know how "sharp" this transition will be. From our experience with a sample having larger thickness of the Si layer, there is a range of temperatures in which both, IKP (weak layering) and BC (strong layering) matching fields coexist.

Changes made: Mo and Si have been labeled in the sketches in Fig. 2. Two sentences have been added before Discussion.

14. Point:

The authors imply that there is a regime where all the layers contribute to the emitted radiation. Are they saying the radiation of the layers buried deep in the multilayer will “shine through” the layers above? What happened to the London penetration depth?

Answer and changes made:

We have reconsidered our interpretation. Please refer to our answer to point 1.

15. Point:

The phrase “as beatings of the f_0 - and $2f_0$ -waves, that suggests an incoherent interference of the microwave emission related to the 50A and 100A periodic length scales” bothers me, particularly the term “incoherent”. Do the authors simply mean these periods produce out-of-phase contributions?

Answer and changes made: Yes, we have rephrased this sentence.

16. Point:

I think the type of structure the authors consider was first treated theoretically in: Critical fields of weakly coupled superconductors, G. Deutscher and O. Entin-Wohlman, Phys. Rev. B17, 1249 (1978). Maybe they could refer to that.

Answer and changes made:

We have added this reference at the end of the last but one paragraph in Supplementary Materials.

17. Point:

One can ask the following technical questions in the Methods section.

Point a.

Amplitude measurements at different frequencies (as required when comparing the amplitude of the harmonics) is always tricky; it is not that clear how (or if) this is done. The coupling strength of the pick up loop to the sample and to the transmission line will surely have a frequency dependence (that for the transmission line can be corrected for).

Answer a:

The frequency dependence of the coupling strength of the sample-antenna-transmission-line system has been corrected for the entire system using the flux-flow oscillator itself. Namely, the amplitude of the peak at the first harmonics at given T and H values was recorded as a function of the dc current in the range of vortex velocities between 20 and 250 m/s, i.e. by a factor of two higher velocity than those dealt with in the paper. This range of vortex velocities corresponds to the viscous flux flow and the CVC of the sample fits to a straight line. Accordingly, unevennesses in the recorded frequency dependence in this regime are related to spurious resonances/reflections in the sample-antenna-transmission-line system. The frequency dependence of the detected amplitude thus recorded was saved in the analyzer as a reference floor for the presentation of the emission peaks at higher harmonics. The frequency dependence of the picked up amplitude has been revealed to be independent on temperature and magnetic field in the ranges 1.8 K to 3.6 K and 3 T to 4.5 T, respectively. This has allowed to correct for the frequency dependence of the coupling strength of the pick-up loop to the sample in the frequency range between 5 GHz and 50 GHz.

Changes made a:

The correction procedure as described above has been added at the end of the Methods section.

Point b:

The low noise preamplifier with a gain of 36 dB. Is that the Lambda unit?

Answer a and changes made:

Yes, RF-Lambda RLNA00M54GA, 0.01 – 54 GHz.

No changes have been made, as the unit name was stated in the first version of the manuscript.

Point c:

I do not understand the sentence: "The signals emitted at different temperatures were further normalized by using a directional coupler with an attenuation of -15 dB at $T = 1.8$ K, -12 dB at 3.0 K, and -1.5 dB at 3.6 K." A typical microwave directional coupler does not have a temperature dependence. What do they mean here?

Answer c:

The referee is right. The directional coupler is not relevant for the present results (at early stages of the experiment we used a modulated locked-in detection scheme to improve the sensitivity, as we did not know the range of emitted powers and expected them to be extremely low). Later on - in the experiment reported in the paper - we used attenuators. The attenuation levels were chosen manually for measurements at different temperatures.

Changes made:

We have rephrased and amended this sentence.

Reviewer #3 (Remarks to the Author):

18. Point: Note that the EM emission due to the Josephson vortex flow was detected in BSCCO in Bae et al. Phys. Rev. Lett. 98, 027002 (2007).

Answer:

Thank you for bringing this paper to our attention.

Changes made:

We have included it in Introduction.

Finally some minor technical remarks:

Point 1):

In p2, the unit of magnetic flux should be Wb.

Changes made:

It is now corrected.

Point 2):

In p3, the symbols in η_J are not defined, although they are later defined in the discussion section. It is better to define them right after η_J .

Changes made:

It is now defined.

Point 3) (Optional):

One advantage to generate EM waves using the vortex lattice is that the frequency can be tuned continuously by magnetic field and current. However, the radiation power is weak (of the order of picowatt). This is probably due to the triangular lattice of Abrikosov vortex, where the radiation from different parts of the system is out-of-phase and interfere destructively. It is helpful to discuss how to enhance the radiation by achieving a rectangular arrangement of vortices.

Changes made:

We have added a very short comment on this before Conclusion.

REVIEWERS' COMMENTS:

Reviewer #1 (Remarks to the Author):

The authors of manuscript NCOMMS-18-13973A have made extensive changes in response to the review comments and thereby considerably improved their manuscript. They have addressed all of my comments except for two points that may need further attention: In response to Point 1 the authors introduce a qualitative model that accounts for the presence of the fundamental and second harmonic in the emission data at 3.15 T and 1.8 K. This model is based on the observation that even though vortices occupy only every second row, current patterns flow in every row representing the underlying sample periodicity and giving rise to double the frequency. This may account for the data; however, naively one would expect that the highest currents flow close to the vortices and that therefore the fundamental should carry high emission power and the second harmonic low power; just opposite to what is shown in Fig. 2a. In regards to point 6, there is a sign error in the equation on page 5 (revised manuscript) describing the IV-curves shown in Fig. 3a. Since $I > I^*$, this equation as written would predict a negative vortex velocity.

Reviewer #3 (Remarks to the Author):

The authors have adequately addressed all the concerns raised by the referees. I think the current version of the manuscript is ready for publication.

Replies to Reviewer #1:

Reviewer #1 (Remarks to the Author):

Point 1:

In response to Point 1 the authors introduce a qualitative model that accounts for the presence of the fundamental and second harmonic in the emission data at 3.15 T and 1.8 K. This model is based on the observation that even though vortices occupy only every second row, current patterns flow in every row representing the underlying sample periodicity and giving rise to double the frequency. This may account for the data; however, naively one would expect that the highest currents flow close to the vortices and that therefore the fundamental should carry high emission power and the second harmonic low power; just opposite to what is shown in Fig. 2a.

Answer:

We agree that the highest currents flow close to the vortices. However, when the vortices cross the layers, the regions of highest and lowest currents are interchanged with the frequency related to the intrinsic sample periodicity, leading to a stronger contribution at the doubled frequency. We believe that our observations will stimulate the development of theoretical models for a quantitative analysis of the spectrum of the emission from the vortex motion in superlattices.

Changes made:

No changes have been made since in the previous version of the manuscript we have already stated that there is no theory available for a rigorous comparison of the emission spectra from the moving vortices in superlattices.

Point 2:

In regards to point 6, there is a sign error in the equation on page 5 (revised manuscript) describing the IV-curves shown in Fig. 3a. Since $I > I^*$, this equation as written would predict a negative vortex velocity.

Answer 2:

You are right. Thank you!

Changes made:

The sign has been corrected.